# Biosurfactant from *Bacillus subtilis* DS03: Properties and Application in Cleaning Out Place System in a Pilot Sausages Processing

**DOI:** 10.3390/microorganisms10081518

**Published:** 2022-07-27

**Authors:** Iana Cruz Mendoza, Mirian Villavicencio-Vasquez, Paola Aguayo, Diana Coello Montoya, Luis Plaza, María Romero-Peña, Ana M. Marqués, Jonathan Coronel-León

**Affiliations:** 1Escuela Superior Politécnica del Litoral, ESPOL, Facultad de Ingeniería Mecánica y Ciencias de la Producción, Campus Gustavo Galindo, Km 30.5, Vía Perimetral, Guayaquil P.O. Box 09-01-5863, Ecuador; ialucruz@espol.edu.ec (I.C.M.); psaguayo@espol.edu.ec (P.A.); dscoello@espol.edu.ec (D.C.M.); laplaza@espol.edu.ec (L.P.); mfromero@espol.edu.ec (M.R.-P.); 2Escuela Superior Politécnica del Litoral, ESPOL, Centro de Investigaciones Biotecnológicas del Ecuador (CIBE), Campus Gustavo Galindo, Km 30.5, Via Perimetral, Guayaquil P.O. Box 09-01-5863, Ecuador; mirvilla@espol.edu.ec; 3Unitat de Microbiología, Facultat de Farmacia, Universitat de Barcelona, 08035 Barcelona, Spain; ammarques@ub.edu

**Keywords:** crude extract biosurfactant, antibiofilm, disinfection, cleaning

## Abstract

Biosurfactants (BS) are amphiphilic molecules that align at the interface reducing the surface tension. BS production is developed as an alternative to synthetic surfactants because they are biodegradable, with low toxicity and high specificity. BS are versatile, and this research proposes using a biosurfactant crude extract (BCE) as part of cleaning products. This paper reported the BCE production from *Bacillus subtilis* DS03 using a medium with molasses. The BCE product was characterized by different physical and chemical tests under a wide pH range, high temperatures, and emulsifying properties showing successful results. The water surface tension of 72 mN/m was reduced to 34 mN/m with BCE, achieving a critical micelle concentration at 24.66 ppm. BCE was also applied to polystyrene surface as pre-treatment to avoid microbial biofilm development, showing inhibition in more than 90% of *Escherichia coli*, *Staphylococcus aureus*, and *Listeria monocytogenes* above 2000 ppm BCE. The test continued using BCE as post-treatment to remove biofilms, reporting a significant reduction of 50.10% *Escherichia coli*, 55.77% *Staphylococcus aureus*, and 59.44% *Listeria monocytogenes* in a concentration higher than 250 ppm BCE. Finally, a comparison experiment was performed between sodium lauryl ether sulfate (SLES) and BCE (included in commercial formulation), reporting an efficient reduction with the mixtures. The results suggested that BCE is a promising ingredient for cleaning formulations with applications in industrial food applications.

## 1. Introduction

The cleaning process in the food industry is a fundamental stage that allows companies to ensure safe and suitable food products for human consumption. Unfortunately, these processes require large volumes of water and chemicals, which have proven unsustainable and have become obsolete processes that do not meet the current needs of the industry [1]. Moreover, wet industrial surfaces or poor cleaning can provide a suitable condition for microbial contamination in the food chain and the development and persistence of ecosystems called *biofilms* that contain pathogenic microorganisms [2]. The biofilm-producing microorganisms that have been detected are *Pseudomonas* spp., *Listeria* spp., *Enterobacter*, *Flavobacterium*, *Alcaligenes*, *Staphylococcus* spp., and *Bacillus* spp. [3]. The food industries remove biofilms with diverse methods, including biological techniques (e.g., bacteriophages, anti-biofilms enzymes, lactic acid bacteria bacteriocins) [4], biosurfactants (e.g., lipopeptides) [5], and physical techniques (e.g., plasma) [6]. BS are new, friendly ecological compounds gaining interest in the industry and academia as an alternative to the traditional synthetic surfactants [7]. 

The main action of surfactants or BS is to reduce the surface tension; therefore, their uses are widespread within various industries as emulsifiers, cleaning agents, lubricants, and stabilizers [8]. Regarding chemical surfactants, the most widely used are anionic ones, such as linear alkylbenzene sulfonate (LAS) and sodium lauryl ether sulfate (SLES) [9], [10]. There is scientific evidence indicating that LAS damages the gills of fish and causes excessive mucus secretion, producing marked alterations in the morphology of amphibian skin, causing hyperplasia or hypertrophy of the epithelium [11]. Likewise, SLES is fundamental in wastewater treatment; however, in high accumulation, it tends to be toxic to microorganisms due to its binding capability with enzymes, structural proteins, and phospholipids, which changes the bacteria cell hydrophobicity [10].

Thus, the problems associated with chemical surfactants motivate the search for efficient and environmentally friendly alternatives. BS are molecules that have shown exceptional surfactant and emulsifying activity, classified by use, properties, and chemical structure [12]. BS have a low critical micellar concentration (CMC) and high stability at extreme pH, temperature, salt concentrations, and biological activity [13]. Similarly, BS have shown low toxicity or irritation, good biodegradability, and even antibacterial capacity [9]. Hence, BS are presented as a suitable option to replace chemical surfactants. The BS can be synthesized from *Bacillus* spp., *Pseudomonas* spp., *Rhodococcus* spp., and *Candida* spp. An influential BS group is lipopeptides (LP), produced mainly by the genus *Bacillus*, being the most effective and efficient, with great interest in the therapeutic, cosmetic, and agri-food industry [14]. The producer strain *Bacillus subtilis* is also Generally Recognized as Safe (GRAS) by the FDA, which supports the use of its byproducts as safe and allows them to be used in direct contact with consumers and the food [15].

The LP surfactin produced by *Bacillus* species is used for biofilm control because it reduces the adhesion of pathogenic bacteria to food in contact with polypropylene [16,17]. For instance, Coronel-León et al. [18] demonstrated that the LP produced by *Bacillus licheniformis* prevented the adherence of pathogenic microorganisms on polypropylene surfaces. Despite the excellent advantages and properties offered by BS, their commercial use is currently limited mainly due to problems related to production costs [19]. Thus, strategies considering agro-industrial food waste and by-products have been promoted [20]. Another important aspect is related to the purity of the biologically based compound. In this context, the pharmaceutical industry requires purity of the compounds greater than 90%, while for agri-food applications, the ranges can vary between 50–60% [21]. Therefore, this research aims to report the production of a biosurfactant crude extract (BCE) from *Bacillus subtilis* DS03, using an economical medium based on an agro-industrial by-product. Then, BCE’s physical and chemical characteristics and their stability under different operative conditions will be evaluated. In addition, the biological activity against *Escherichia coli* ATCC 11775, *Staphylococcus aureus* ATCC 12600, and *Listeria monocytogenes* ATCC 19115 were studied. Finally, the BCE was evaluated as a potential sanitizer in open cleaning systems in the meat processing laboratory. 

## 2. Materials and Methods

### 2.1. Materials

The reagents: dextrose, yeast extract, sodium nitrate (NaNO_3_), disodium phosphate (Na_2_HPO_4_), monopotassium phosphate (KH_2_PO_4_), magnesium sulfate (MgSO_4_ 7H_2_O), ferrous sulfate heptahydrate (FeSO_4_ 7H_2_O), calcium chloride (CaCl_2_), Müller-Hinton broth (MHB), tryptic soy agar (TSA), peptone water, and crystal violet were purchased from Merck (Merck, Germany). The analytical grade solvents: chloroform (CHCl₃), ethyl acetate (C_4_H_8_O_2_), methanol (CH_3_OH), acetic acid (33%, CH_3_COOH), and hydrochloric acid (37%, HCl) were obtained from Fisher Chemicals (Fisher Scientific, United States). The components used to prepare the cleaning product COP-A were sodium hydroxide (NaOH), sodium gluconate (C_6_H_11_NaO_7_), sodium tripolyphosphate (Na_5_P_3_O_10_), sodium lauryl ether sulfate (SLES), and demineralized water; all were given by Chemicals Clean company (Guayaquil, Ecuador). Avocado oil (68% linoleic acid), soybean oil (62% oleic acid), and coconut oil (45% palmitic acid) were provided by a local commercial company, namely Laboratory Cevallos (Guayaquil, Ecuador). Molasses (sugar cane), whey, and cassava starch were donated by local industries (Molasses Tababuela, Dairy Industry Rey Lacteos, and La Pradera, respectively). The microorganisms *Escherichia coli* ATCC,11775, *Staphylococcus aureus* ATCC 12600, *and Listeria monocytogenes* ATCC 19115 were acquired from Microbiologics (St. Cloud, MN, USA).

### 2.2. Production of the Biosurfactant Crude Extract

*Bacillus subtilis* DS03 strain was isolated from cocoa crops, producing the microbial surfactant [22]. The medium used for BS production was MMI (20 g/L dextrose, 1 g/L yeast extract, 8.5 g/L NaNO_3_, 5.7 g/L Na_2_HPO_4_, 4 g/L KH_2_PO_4_, 0.21 g/L MgSO_4_ 7H_2_O, 0.01 g/L FeSO_4_ 7H_2_O, and 7 × 10^−6^ g/L CaCl_2_). Three alternative carbon sources were evaluated for BCE production: molasses, whey, and cassava starch at 20 g/L, similar to dextrose concentration in the MMI culture medium. For the culture media preparation, the sources of carbon, nitrogen, and mineral salts were autoclaved (JP Selecta 4001759, Barcelona, Spain) for 15 min at 121 °C. Flasks (Pyrex N°4980, VWR, Radnor, PA, USA) with 400 mL of the culture media were inoculated with 2% *v*/*v* of bacterial suspension (turbidity 2 according to the McFarland standard) and incubated with shaking at 110 rpm (New Brunswick INNOVA 44R; Alemania, Hamburg) for 24 h at 37 °C. 

Before removing the BCE, the bacterial cells were removed from the culture medium by centrifugation at 8000× *g* for 15 min at 4 °C (Avanti TM j-20 centrifuge; Bekman Coulter, Indianapolis, IN, USA). Then, the cell-free supernatants were subjected to acid precipitation using concentrated HCl until reaching pH 2 and were left overnight at 4 °C in a sealed tube (Ultracruz, Santa Cruz, CA, USA). The BCE was collected by centrifugation for 20 min at 4 °C, 11,000× *g*, and washed twice with acid distilled water at pH 2 to eliminate any impurities. The BCE was dried in an oven (Thermo Scientific 6963, Waltham, MA, USA) at 40 °C until obtaining a constant weight and then quantified by gravimetry (g/L) [22]. 

### 2.3. Evaluation of Physical-Chemical Properties of BCE

BCE properties were compared with the biosurfactant organic extract (BOE) fraction. BOE was obtained from the BCE extraction with methanol (1:2 p/v). The organic phases were passed over anhydrous sodium sulfate, concentrated in a rotary vacuum evaporator (Büchi, Flawil, Switzerland), and weighed. BOE was chromatographed on a silica gel column. Elution was performed with chloroform/methanol/ammonium hydroxide (65:35:5A). Fractions were collected in vials and monitored by thin-layer chromatography (TLC) with the same elution solvent. Then, fractions were revealed by ninhydrin (Sigma, St. Louis, MO, USA) specific for amino acid moiety. Finally, samples that showed the presence of amino acid and the same retention factor (RF) were grouped and analyzed by surface tension measurements [23].

**BS stability at different pH and temperature.** BCE and BOE stability was evaluated against different pH and temperature conditions. Solutions of BOE (620 ppm) and BCE (500 ppm) in 20 mL of distilled water were prepared. Then, the surface activity was measured in a K6 tensiometer (Krüss, Hamburg, Germany) by subjecting the suspensions to temperatures 4, 16, 26, 50, 60, 67, 75, 80, 90, 100, and 121 °C for one hour. Likewise, the solution was adjusted to pH values 2, 3, 5, 7, 10, 11, and 13, using HCl and NaOH.

**Emulsifying activity.** Similarly for BCE and BOE, the emulsifying activity was evaluated according to the methodology described by Coronel-León et al. [23]. The evaluated oils were avocado, soybean, and coconut. The tubes were shaken using a vortex (Vortex-Genie, Bohemia, NY, USA) for 2 min and allowed to stand for 24 h. The emulsification index (E_24_) was calculated by the ratio between the height (mm) of the generated emulsion and the total height (mm) of the mixture, multiplied by 100. Distilled water was used as a control. The emulsion test was performed using a 100 ppm BS aqueous solution. 

**Critical micellar concentration (CMC).** The CMC of the BCE and BOE was determined from a stock dilution of 620 and 500 ppm of each compound, a Krüss K6 tensiometer (Krüss, Hamburg, Germany), using the ring method. The different serial concentrations were prepared in distilled water at pH 7 in a final volume of 20 mL. Subsequently, the surface tension was measured for each solution at 25 °C. Finally, the data is represented graphically, surface tension versus the different concentrations. The intersection point from the slope of the two distinct portions of the graph was used to find the CMC, beyond which no changes in surface tension were observed [24]. 

### 2.4. Anti-Adhesion Assay on Polystyrene Surface

The BCE pre-treatment action was evaluated considering three microorganism models, *E. coli* ATCC 11775, *S. aureus* ATCC 12600, and *L. monocytogenes* ATCC 19115, that present problems of biofilm formation in meat industries. Previously, the 96-well microtiter microplates (Brand Plates, Wertheim, Germany) were filled with 200 µL of several BCE concentrations (2–4000 mg/L). Then, the microorganisms were inoculated in conditions to form biofilm using the method stated by Coronel-León et al. [18]. Then, for the quantification of microbial adhesion inhibition, wells were washed with distilled water, fixed for 15 min with methanol, and stained for 20 min with 1 wt.% crystal violet. After washing and drying, the stain in the wells was diluted with 200 µL acetic acid, and the absorbance was determined at 595 nm. Percentages of microbial adhesion inhibition were calculated using Equation (1): Percentage of microbial adhesion inhibition = [1 − (Ac/Ao)] × 100(1)
where Ac represents the absorbance of the well with BCE at concentration c, and Ao is the absorbance of the control well (absence of BCE). For the post-treatment with BCE, the attached microbial cells were put in contact with 200 µL of different concentrations of BCE (2–4000 mg/L) and incubated at 25 °C for 6 h. The quantification was performed as in the pre-treatment procedure.

### 2.5. Antimicrobial Activity of BCE: Time-Kill Curve

Suspensions of microorganisms with a cell density of about 1 × 10^8^ CFU/mL were prepared in buffered peptone water from an overnight culture of each strain—*E. coli* ATCC 11775, *S. aureus* ATCC 12600, and *L. monocytogenes* ATCC 19115 on TSA at 30 °C. To evaluate, the antimicrobial activity over BCE was added to a final concentration of 800 ppm. The contact between BCE and microorganisms was monitored for 60 min, and viable cell counts calculated the bacterial viability in CFU/mL after exposure. Briefly, after ten-fold serial dilution of treated bacterial suspension, 0.1 mL aliquots were spread on the TSA plate surface and incubated at 37 °C for 24–48 h. Control bacterial suspensions were performed in parallel. Viability was calculated with Equation (2), where N is the average number of colonies counted on three plates, and d is the dilution factor [25].
Number of bacteria = N/d(2)

### 2.6. Design of the Cleaning Out-of-Place System (COP) Using BCE

The potential of BCE as a cleaning product was evaluated by designing an open cleaning system in the meat processing laboratory facilities at the ESPOL pilot plant after fine sausage production. The sausage formulation is shown in Table 1. Firstly, the critical equipment and utensils—knives, mill, cutter, stuffer, and slicer—were established. Secondly, in collaboration with the Chemical Cleans company, three products were formulated to be used as cleaning agents after the sausage production process. The first product is a reference compound, namely COP-A, described previously. For the product COP-B, the SLES was replaced with 5% *v*/*v* BCE, maintaining the rest of the ingredient’s constant (sodium gluconate (C_6_H_11_NaO_7_), sodium tripolyphosphate (Na_5_P_3_O_10_) and demineralized water). Finally, COP-C product contained BCE (5%) and demineralized water. The products were evaluated as part of a COP process at 3% *v*/*v* for seven minutes recommended by Chemicals Clean. 

#### 2.6.1. Description of the COP Cleaning Method

Once the products and the application conditions were established, the cleaning process was performed, beginning with disassembling the moving parts in the equipment and utensils described earlier. Subsequently, the organic residues were swept, and the initial rinsing at room temperature was performed. Next, the cleaning solution was applied, and mechanical scrubbing was performed with sponges and brushes. Finally, the final rinse was applied to remove meat and detergent remains. Subsequently, the microbiological verification was performed. 

#### 2.6.2. Microbiological Verification

The samples were collected to perform the microbiological analyses according to the methods proposed by normative regional N° 461-2007/MINSA. The samples were taken after sausage production for the initial microbial load and after the cleaning procedures with the different cleaning products. Samples after cleaning were randomly taken from various spots on the surfaces. Each sample was taken by swabbing 100 cm^2^ of the most critical part of the equipment and utensils—knives, mill, cutter, stuffer, and slicer. The swabs with the samples were placed in 10 mL peptone water and mixed for two minutes to release the microorganisms present. To assess the efficiency of the COP process, viable mesophilic aerobic microorganisms and total coliforms were quantified using rapid counting methods (Compact Dry Nissui). The results were expressed as log colony forming units (CFU)/cm^2^. In addition, microbial load reduction was calculated using Equation (3), where *Ci* is the average value of the initial sample before treatment and *Cf_m_* is the average value of the final sample after treatment, both reported in log CFU/cm^2^.
% Microbial load reduction = [(*Ci* − *Cf_m_*)/*Ci*] × 100(3)

### 2.7. Statistics

Statistical analyses were performed using Statgraphics^®^ Centurion XVI software statistics (StatPoint Technologies, Warrenton, VA, USA). Descriptive statistics were used to observe trends in the dataset, and all the results were represented as the average of three independent experiments with their correspondent standard deviation. An analysis of variance (ANOVA) was performed with a 95% confidence interval to evaluate the microbial load reduction in surfaces using the different solutions. The Least Significant Difference (LSD) Fisher test was performed at a 5% significance level to analyze which groups could present significant differences.

## 3. Results and Discussion

### 3.1. Production of Biosurfactant

In previous work, we isolated, identified, and selected *B. subtillis* DS03 from cocoa crops for producing an organic extract with high surface activity. A preliminary evaluation showed that BCE production was affected by carbon source (e.g., C_6_H_12_O_6_), nitrogen source (e.g., NaNO_3_), and phosphate salts (e.g., Na_2_HPO_4_ and KH_2_PO_4_) [22]. This study evaluated three alternative carbon sources for BCE economic production (i.e., cassava starch, whey, and molasses) according to a new sustainable approach to the biotechnology process. All alternative carbon sources contained monosaccharides (i.e., glucose and fructose), disaccharides (e.g., sucrose and lactose), and polysaccharides (e.g., starch). The effect of different carbon sources on BCE production, surface tension, and biomass are shown in Figure 1. In the medium with cassava starch (glucose as the majority component), the growth of strain DS03 (1.1 g/L) was significantly promoted compared to other mediums. However, the BCE production was limited (0.08 g/L), and the surface tension was the highest (68 ± 1 mN/m), indicating low BS richness in BCE. *Bacillus* DS03 in the medium with whey (lactose) showed a growth of 0.66 g/L and the most insufficient BCE production (0.045 ± 0.1 g/L) of all mediums, with a surface tension of 49 mN/m, which was lower compared to the starch medium. The output with molasses (i.e., rich in sucrose, fructose, and glucose) showed the most significant reduction in surface tension (35 mN/m) than the other mediums, similar to glucose (on average 33.5 ± 1 mN/m). Moreover, molasses in bacterial growth reported a significant rise compared to glucose and whey, and the highest BCE production (0.85 ± 0.05 g/L) was observed even compared to the control (0.618 ± 0.04). These results indicate that disaccharides containing glucose performed better indices (e.g., biomass and BCE) than lactose. The decrease in the surface tension was significant with *Bacillus* DS03 (35 mN/m) compared with a biosurfactant produced by *Lactobacillus* strains (41.90 mN/m) [26] also obtained with molasses. Suitable surfactants are reported to diminish the surface tension to 35 from 72 mN/m [27], indicating *Bacillus* DS03 had effectiveness at the interface. Compared with glucose and other carbon sources, the BCE production increment with molasses was due to the substrate’s significant sugar content, minerals, and vitamins. These outcomes agreed with the efficient production of biosurfactants from *Bacillus subtillis* reported in the literature. However, the relationship between microbial growth and BCE production using a medium that contains molasses as a carbon source, NaNO_3_ as a nitrogen source and phosphate salts (Na_2_HPO_4_ and KH_2_PO_4_), is shown in Figure 2. The lag phase lasted around 9 h, and this is because molasses is a complex substrate. Then, the exponential growth phase started from hour 9 until hour 15; simultaneously, a significant decrease in surface tension was evident (67–53 mN/m). After 27 h, maximum BCE production was reached (0.012–0.152 g/L) while the surface tension decreased to its lowest point, 28.7 mN/m. Since this relationship occurred within the stationary phase, it is suggested that BCE corresponds to a secondary metabolite. However, it is essential to note that the growth trend of DS023 increases again after 27 h; this pattern could be associated with the complex composition of molasses. The behavior found here is similar to that described by Coronel-León et al. [18], where the production of lichenysin was observed from strains of *B. licheniformis*. During the exponential phase (12–36 h), the production of lichenysin increased proportionally with microbial growth and surface tension reduction.

Thus, using agro-industrial waste or by-products as feedstock will reduce the initial costs of raw materials. Molasses has 40–60% wt. of sugars and contributes several amino acids, vitamins, and inorganic salts [28]. For this reason, the significant production results found with *B. subtilis* DS023 are associated with molasses composition. However, this production result is not a general rule. The preferred carbon source utilization may vary among different strains of *Bacillus* spp., which may be attributed to differences at molecular levels, according to Gaur et al. [29]. Likewise, scientific evidence has suggested that the availability of hexose and pentose sugars, such as the ones mentioned above, help determine the pattern growth and concentration of surfactant production by *Bacillus* strains [30]. It has also been demonstrated that high initial sugar concentration helps minimize economic costs. In this term, *Bacillus* can withstand high glucose concentration, hence the possibility of utilizing carbon sources with significant sugar percentage in their composition [31,32]. Molasses is an essential source for obtaining microbial surfactants; however, it is necessary to mention that its use also implies the development of additional strategies for product purification. These techniques include dialysis, microfiltration, and adsorption, which should be further investigated for better separation of the bioproducts from the complex molasses media [33]. However, the purification grade of microbial surfactant (MS) depends on the final application of these compounds. Thus, this research focuses on BCE and its properties described in the following section to develop an economic process.

### 3.2. Physical-Chemical and Biological Properties of BCE

The biosurfactant organic extract (BOE) was obtained using methanol, obtaining a recovery of 0.052 g/L. The BOE purification was also performed using the column chromatography technique from the organic extract, from which three purified fractions (BOE_1_, BOE_2_, and BOE_3_) were obtained and grouped based on their polarity. The purified fractions present values of Rf_BOE1_ = 0.89, Rf_BOE2_ = 0.38, and Rf_BOE3_ = 0.12. Regarding the decrease in control surface tension (70 mN/m), BOE_1_ and BOE_3_ cause a decrease of 45 mN/m and 53 mN/m, respectively. However, the BOE_2_ fraction decreased the surface tension from 70 to 29 mN/m. Therefore, the fraction that reduced surface tension to more than 20 mN/m was BOE_2_, and this fraction was used to compare stability and physical-chemical properties with the BCE product. 

#### 3.2.1. Effect of pH and Temperature on BCE Action

Properties, such as surface tension reduction, are essential for BCE application in cleaning systems; hence, it was assessed at various pH and temperatures. The surface tension value diminution for BCE and BOE_2_ were 34 mN/m and 30 mN/m, respectively. As seen in Figure 3A, the surface tension of BCE and BOE_2_ was maintained over the range of temperature evaluated (4–121 °C). It remained stable up to a sterilization temperature (121 °C). However, Figure 3B shows that as the pH increased from 6 to 14, the surface tension of BCE and BOE_2_ remained stable. In the case of BCE, a significant increase in surface tension (59 mN/m) was observed under acidic conditions (below pH 4). It corroborates the resistance of biosurfactants when maintaining their effectiveness in extreme conditions [34]. For instance, the lipopeptide produced by *Nesterenkonia* sp. MSA31 was found to be thermo-stable in temperature ranges between 4–121 °C and at pH 6.0–9.0. These properties are suitable for ice cream and cosmetic industries [35]. Moreover, microbial lipopeptide isolated from *Pontibacter korlensis* SBK-47 showed stability over a wide pH range (4–10) and temperature up to 100 °C [36]. At the same time, Hentati et al. [37] reported that MS from *Bacillus stratosphericus* FLU5 maintained a steady surface tension (on average 35.5 mN/m) over a broad pH range (2.1–12) and covered an extensive temperature range (4–121 °C). The BS obtained from *B. paralicheniformis* is known to present stability even after autoclaving it at 121 °C for 20 min and maintained in storage for 6 months at −18 °C, showing stable surface activity at pH between 5 and 11 [38]. The observed broad pH range and temperature against surface tension stability of BCE and BOE_2_ suggests good compatibility in diverse industrial applications. With more specific information, Fei et al. [9] mentioned that cleaning products used in households and industries are generally subjected to 25–60 °C and pH 7.0–11.0. Therefore, this suggests that BOE_2_ and BCE are promisingly thermal-stable and pH-tolerant ingredients for cleaning formulations. Similarly, Schultz and Rosado [39] described that cleaner product stability is essential for industrial processes associated with extreme pH, temperature, and ionic strength conditions. 

#### 3.2.2. Emulsifying Activity

When a solution that contained BCE and BOE_2_ (separately) was in contact with different oils, both compounds were found to form stable emulsions with avocado (68% linoleic acid), soybean (62% oleic acid), and coconut (45% palmitic acid) oils. The emulsification index (E_24_) with BCE was above 80%, with all oils experimented. While, for BOE_2_, the emulsification index with avocado, soybean, and coconut oils was 98%, 95%, and 81%, respectively. The emulsification index designates the strength of the surfactant to prevent phase separation [40]. The BCE and BOE_2_ acted on the mixture of immiscible liquids (i.e., aqueous medium/vegetable oil), forming an emulsion in both phases. The formed emulsion showed a complete and homogeneous appearance with the concentration tested. The emulsification rates with BCE and BOE_2_ were more than 80% for the three oils assayed, which showed the higher capacity of both compounds to stabilize emulsions with vegetable oils for 24 h. 

With BCE and BOE_2,_ the avocado and soybean oils formed strong oil/water emulsions, while the mixture with coconut oil presented a soft emulsion in its aqueous phase. It could be attributed to the predominant fatty acids in these oils. Palmitic acid is saturated and has a shorter chain (C16: 0) compared to unsaturated fatty acids, such as oleic (C18:1 *cis*-9) and linoleic (C18:2, *cis*-9,12) acid, predominant in avocado and soybean oils, respectively [41]. Certain studies describe BS as low molecular weight compounds formed mainly by glycolipids and short chains of lipopeptides, unlike bioemulsifiers (BE) with a high molecular weight [42]. The lipopeptides produced by the species *B. subtilis* with low molecular weight suggest better surface activity, reducing surface tension further than emulsifiers [43]. However, the BS produced by strain DS03 proved to have excellent properties in forming emulsions. The emulsifying capacity accompanies the HLB value (i.e., hydrophilic-lipophilic balance), which denotes the ability of a surfactant to form oil-in-water (O/W) or water-in-oil (W/O) emulsions [44]. In this case, the BS ability was successful in forming O/W emulsions. 

#### 3.2.3. Determination of Critical Micellar Concentration (CMC)

The CMC is the concentration at which micelles begin to form. Beyond the CMC, no changes in interfacial properties are noticed [45]. The desired strains produce BS with the lowest CMC values and high surface tension reduction. In this study (Figure 3C,D), a progressive decrease in surface tension can be observed at higher BCE concentrations, reaching a minimum value of 33 mN/m and a CMC of 24.66 ppm. BOE also decreased the surface tension to 30 mN/m and had a CMC of 19.16 ppm. The effectiveness and efficiency can describe BS. Thus, the CMC results suggest that BCE and BOE are efficient due to the low concentration required to achieve maximum reduction in surface tension. The CMC values were also close to the lipopeptide biosurfactants from different *Bacillus* species [46,47]. According to the scientific literature, one characteristic that makes MS superior to synthetic surfactants is the low CMC values [48,49]. For instance, LAS and SLES are anionic compounds used widely in cleaning operations and have a CMC of 100 ppm [50] and 1100 ppm [51], respectively. With both compounds, the concentrations are above BCE and BOE reported here.

According to the results, the BOE and BCE have stable physical–chemical properties comparable to biosurfactants reported by several researchers. In addition, obtaining a BCE requires less time and resources than BOE. Therefore, our research demonstrates the feasibility of using BCE to replace BOE. Thus, the following sections will discuss the biological properties of BCE and its potential to be part of the cleaning-out-place system applied in food processing.

### 3.3. Antiadhesion Assay on Polystyrene Surface

The BCE presented antiadhesive activity against *E. coli* ATCC 11775, *S. aureus* ATCC 12600, and *L. monocytogenes* ATCC 19115. When a polystyrene surface was pre-treated with BCE, an apparent decrease in bacteria adhesion was observed (Figure 4A). The anti-adhesive effect of BCE was concentration-dependent. At the studied concentrations up to 1000 ppm, the highest anti-adhesive effect against *E. coli* reached 91.64%. In contrast, a lower inhibition was obtained for *S. aureus* (56%) and *L. monocytogenes* (71%) at the same concentration. It is important to note that the maximum inhibition for each microorganism tested occurred in a concentration higher than 2000 ppm.

On the contrary, another approach is to disrupt the biofilm formed by microorganisms. Figure 4B shows the obtained data for the post-treatment using BCE, and these anti-adhesion values were lower than those reported in the pre-treatment in all microorganisms tested. The maximum disruption produced by BCE was 50.10% for *E. coli*, 55.77% for *S. aureus*, and 59.44% for *L. monocytogenes* in a concentration higher than 250 ppm. Thus, the action of BCE was more effective in the pre-treatment against the three tested microorganisms. However, in terms of BCE concentration, a low quantity was used post-treatment to achieve maximum inhibition. 

The ability of microorganisms to build biofilm as a shield against disinfection techniques could develop antimicrobial-resistant foodborne pathogens [52]. 

The food industries remove biofilms with diverse methods, including biological techniques (e.g., bacteriophages, anti-biofilms enzymes, lactic acid bacteria bacteriocins) [4], and microbial surfactants (e.g., surfactin and rhamnolipids) [5]. Several researchers have reported that the BS action in biofilm prevention could result from the repulsion forces between the microbial surface’s negative charges and the surface’s negative charge coated with MS molecules [18]. Likewise, the disruption effect results from MS penetration and absorption at the interface between the solid surface and the attached biofilm-forming bacteria, thus reducing the interfacial tension and facilitating biofilm removal [5]. The results obtained here were also reported by Singh et al. [53], which described that lipopeptide (6 g/L) produced by *Bacillus tequilensis* inhibited more than 99% of microorganisms on stainless steel. Moreover, the purified pontifactin showed a maximum anti-adhesive activity of 99% against *Bacillus subtilis*, *Staphylococcus aureus*, *Salmonella* typhi, and *Vibrio cholerae* at 2 mg/L [36].

Similarly, De Araujo et al. [54] evaluated the potential of two biological surfactants against biofilm formation by *L. monocytogenes* and *Pseudomonas fluorescens* on surfaces, such as stainless steel and polystyrene. They reported that rhamnolipids and surfactin exhibited anti-adhesive activity on polystyrene at 79% and 54%, respectively. In contrast, biofilm formation on stainless steel surfaces was reduced to 83% and 73%, respectively. These observations suggest that the BCE finds promising prospects for many cleaning applications. Further, Smith et al. [55] also indicated that using products that contain biosurfactants in conjunction with or as an alternative to heavy chemical cleaners may be more effective in efficient disinfection. 

### 3.4. Evaluation of Antimicrobial Activity

Three microorganisms, particularly problematic for the food industry, were chosen to analyze the antimicrobial activity of the BCE. When bacterial cells were treated at 800 ppm of BCE, an effective reduction in viability was found after 15 min of treatment, achieving a maximum reduction (7 log CFU/mL) after 60 min (Figure 5). The microbial decline of *E. coli* and *L. monocytogenes* has a similar pattern. In the first 15 min of contact with the BCE, there is an approximate decrease from 8 to 4 log CFU/mL. The behavior of both microorganisms (Figure 5A,C) showed a similar reduction throughout the 60 min, where the decrease was 7 log CFU/mL.

Similarly, *S. aureus* (Figure 5B) achieved the same microbial reduction rate at 60 min of contact. However, it did not show such a drastic decrease in the first 15 min, but it decreased by 6 log CFU/mL at 30 min. When comparing results with the control, it can be determined that a microbial reduction reached approximately 90% at 60 min of contact regardless of the microorganism under study. The only variation is the speed of action, which may depend on the specific characteristics of each microorganism and how they interact with the BCE. However, the effect was similar for *E. coli* and *L. monocytogenes*, regardless of their Gram-negative or Gram-positive nature. They showed the same behavior pattern demonstrating that the BCE compound has a significant inhibitory effect that allows the elimination of up to 90% of the microorganisms in direct contact with it in approximately 60 min. Several factors can affect the antimicrobial efficacy of biocides, but contact time is probably the most critical factor. A similar reduction (more than 90%) after 1 h of growth has been reported in Gram-positive and Gram-negative bacteria treated with lipopeptide biosurfactant (6000 ppm) produced by *Bacillus tequilensis* strain SDS21 [53]. Coronel-León et al. [25] also reported a reduction in viability of 6 log units when lichenysin/cationic (i.e., amino acid surfactant mixture) was used against *E. coli* O157:H7 and *L. monocytogenes.* Consequently, the pattern obtained here is consistent with the action of conventional lipopeptides. The antimicrobial effect improved as the contact time increased, suggesting that time might be essential to eliminating the microbial population. 

### 3.5. Evaluation of BCE in a Cleaning-Out-Of-Place (COP) Process

From the qualitative analysis of the sausages process, grinding, cutting, and sausage stages were determined as critical points due to the greater contact time between raw meat and the equipment and utensils used. For this reason, the evaluation of cleaning products was focused on these stages, representing a higher microbial risk for the product and consumer. The knives, mill, cutter, stuffer, and slicer used in sausage production presented an initial count of aerobic mesophilic microorganisms (Figure 6A) range of 4.2–4.8 log CFU/cm^2^. Likewise, the initial count of total coliforms (Figure 7A) range was 3.5–4.6 log CFU/cm^2^. The microbial profile of knives, mill, cutter, stuffer, and slicer used in the sausage process treated with the three products, COP-A, COP-B, and COP-C, are shown in Figure 6 and Figure 7. A reduction in aerobic mesophilic microorganisms (Figure 6B–D) and total coliforms (Figure 7B–D) was observed with COP-A, COP-B, and COP-C products. 

Regarding aerobic mesophilic microorganisms, after cleaning operation using a conventional product COP-A (Figure 6B), the counts were significantly reduced, reporting a range of 0.15–1.25 log CFU/cm^2^ without detection in the slicer. Similarly, the COP-B product (Figure 6C) triggered a reduction in the count of these microorganisms to a range of 0.89–1.10 log CFU/cm^2^. No aerobic mesophilic microorganisms were detected in the slicer. It is essential to mention that the microbial reduction in knives (from 4.62 to 2.64 log), mill (from 4.42 to 2.30 log), cutter (from 4.35 to 2.31 log), stuffer (from 4.84 to 2.17 log), and slicer (from 4.28 to 1.83 log) using COP-C (Figure 6D) as a cleaner product were lower than those found for conventional treatment COP-A and novel product COP-B. Based on the American Public Health Association (APHA) [56], the limit established for aerobic mesophilic microorganisms was 2.00 log CFU/cm^2^. Above this limit indicates unsatisfactory hygienic–sanitary conditions. Therefore, our results report that aerobic plate counts from products COP-A and COP-B were lower than the limit for the surface evaluated, whereas plate count from the COP-C process in all equipment and utensils did not achieve hygienic conditions. The ANOVA test showed significant differences between the three products (*p* < 0.05), but the LSD Fisher test demonstrated that COP-A and COP B had no significant differences, contrary to COP-C, which had differences with each product mentioned before. These results support the quantitative analysis where COP-C showed less efficacy in reducing aerobic mesophilic microorganisms on the tested surfaces. Finally, it has been demonstrated that COP-A and COP-B have a similar effect on this type of microorganism and can be used as cleaning agents on surfaces in the food industry. 

However, with total coliforms after cleaning operation using a conventional product COP-A (Figure 7B), the counts were highly reduced in a range of 0.39–1.41 log CFU/cm^2^, without detection in stuffer and slicer. Likewise, COP-B product (Figure 7C) triggered a reduction in the count of these microorganisms in a range of 1.15–1.29 log CFU/cm^2^ and was not detected in knives, mills, cutter, stuffer, and slicer. Regarding COP-C product (Figure 7D), the counts decreased on average by 2.55 log CFU/cm^2^ for all equipment and utensils. COP-C as a cleaner product was less effective than COP-A and COP-B products. Based on Sneed’s recommendations [57], the limit established for total coliforms is 1.00 log CFU/cm^2^; therefore, COP-A achieved satisfactory hygienic conditions for the mill, stuffer, and slicer. In comparison, knives and cutters presented counts higher than the recommended limit. Otherwise, with COP-B, the cleaning standard was met only in the slicer. Despite the decrease in coliforms count with COP-C, the reduction was unsatisfactory. The ANOVA and LSD Fisher test demonstrated that COP-A and COP-B had no significant differences, contrary to COP-C, which had differences with each product mentioned before, related to the aerobic mesophilic microorganisms’ results. 

As mentioned, COP-A and COP-B products efficiently decreased the aerobic mesophilic microorganisms in all equipment and utensils tested. However, the action of these compounds against total coliforms depends on the surface evaluated. For instance, COP-C did not achieve both microbiological limits. The results of microbial reduction load showed an apparent decrease in initial microbial load with all products in all equipment and utensils used (Table 2). With COP-A, the average microbial reduction was 83.14% and 85.66% for aerobic mesophilic microorganisms and total coliforms, respectively. In the same line, COP-B caused a decrease of 82.24% and 75.59% in aerobic mesophilic microorganisms and total coliforms, respectively. Finally, the averages of microbial reduction with COP-C were around 50.07% (aerobic mesophilic microorganisms) and 35.58% (total coliforms). The overall microbial reduction suggests that the COP-A, COP-B, and COP-C action could be improved by applying prevention strategies to reduce the initial microbial load. A preventive approach could be the correct and strict application of good manufacturing practices (GMPs) to decrease the microbial contamination risk. The combination with disinfection procedures can also help, for example, the use of physical operations (e.g., ultrasound, plasma technology, ultraviolet light) or biological products (e.g., organic acids, essential oils, enzymes). 

The decreased count of microorganisms in equipment and utensils used during sausage production might be caused by a detergent-like mechanism produced by the BCE (COP-B and COP-C) related to the surface activity. BS penetrate biological membranes and alter their structure and function, which can be due to either permeabilization or the solubilization of the membrane [58,59]. The COP-B effect is comparable with Brasil et al. [60], in which a significant decrease (2 log CFU/cm^2^) in the number of pathogenic and spoiling microorganisms in knives applying ultrasound with potable chlorinated water were found. 

## 4. Conclusions

BCE was successfully obtained using a circular economy approach with molasses as a carbon source. This compound reported surface activity by reducing the interfacial tension and showing low concentration to saturate the medium. Furthermore, emulsifying abilities were observed in O/W emulsions; however, further research in characterization and testing will continue. Its stability under significant operations conditions allows BCE to be used in the cleaning process for different industries. The antimicrobial and antibiofilm activities are comparable with purified microbial surfactants reported in the scientific literature. The formulas with the BCE incorporation showed positive results in sausage processing. Cleaning of knives, mills, cutters, stuffers, and the slicer using COP-B (BCE 5% *v*/*v*) promoted a significant decrease in the number of aerobic mesophilic microorganisms and total coliforms. Therefore, this research proposes to use semi-purified BS as an excellent opportunity to develop an economical process for diverse applications. In addition, formulating a biological/chemical products strategy could reduce environmental impacts.

## Figures and Tables

**Figure 1 microorganisms-10-01518-f001:**
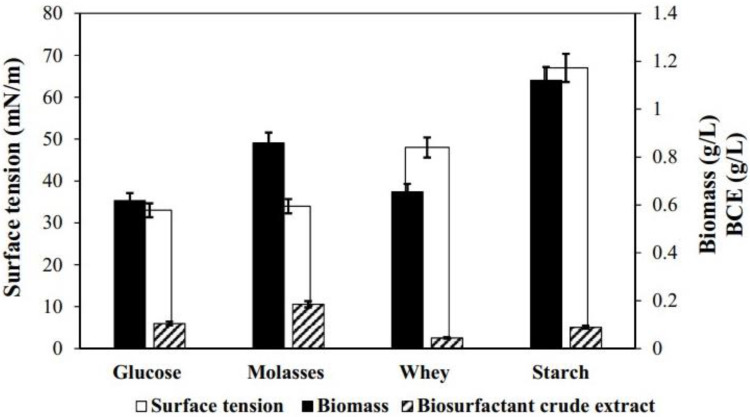
Surface tension (white bars), biomass (black bars), and biosurfactant crude extract (BCE, gradient-filled bars) production in glucose, molasses, whey, and starch by *Bacillus subtilis* DS03 growth. Note the difference in y-axis scales between surface tension against biomass and BCE.

**Figure 2 microorganisms-10-01518-f002:**
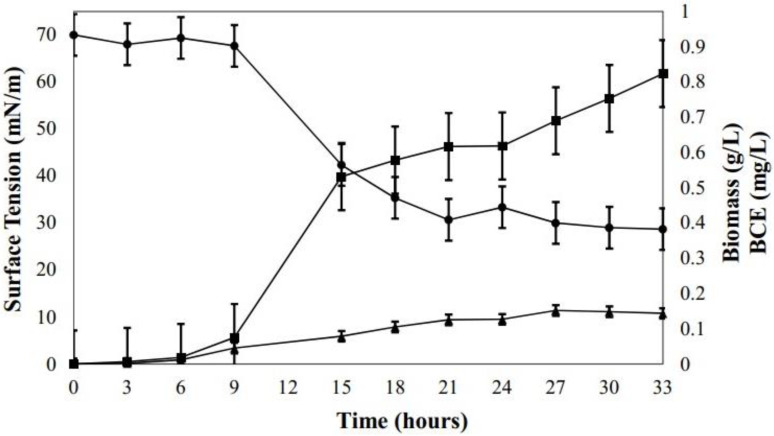
Time course of biosurfactant crude extract (BCE, ▲) production, surface tension (●), biomass (■) production in molasses medium by *B. subtilis* DS03. Note the difference in y-axis scales between surface tension against biomass and BCE.

**Figure 3 microorganisms-10-01518-f003:**
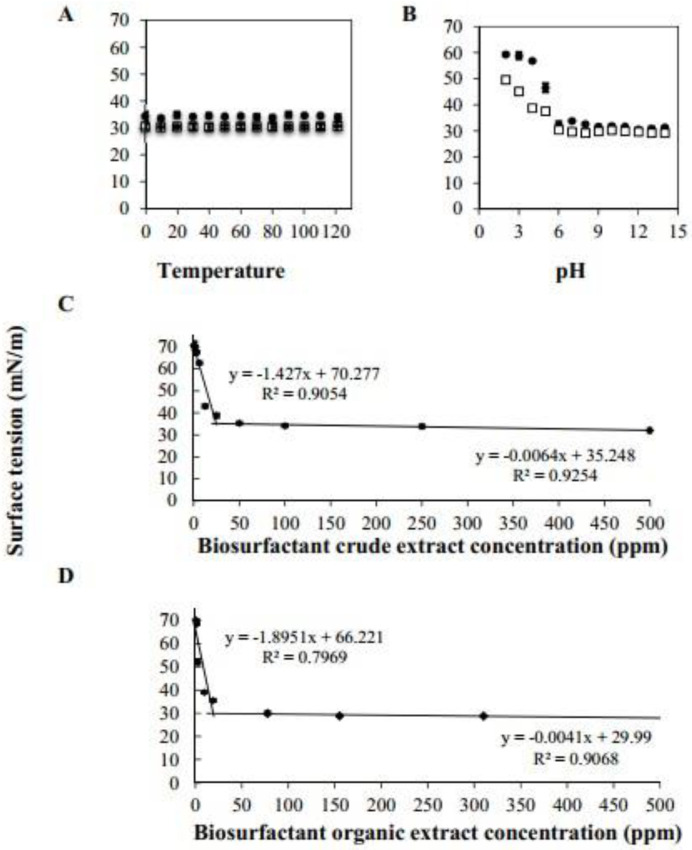
Surface tension against temperature (**A**), pH (**B**) for biosurfactant crude extract concentr− ation (BCE, white square) and biosurfactant organic extract concentration (BOE, ●). Critical micelle concentration of BCE (**C**) and BOE (**D**).

**Figure 4 microorganisms-10-01518-f004:**
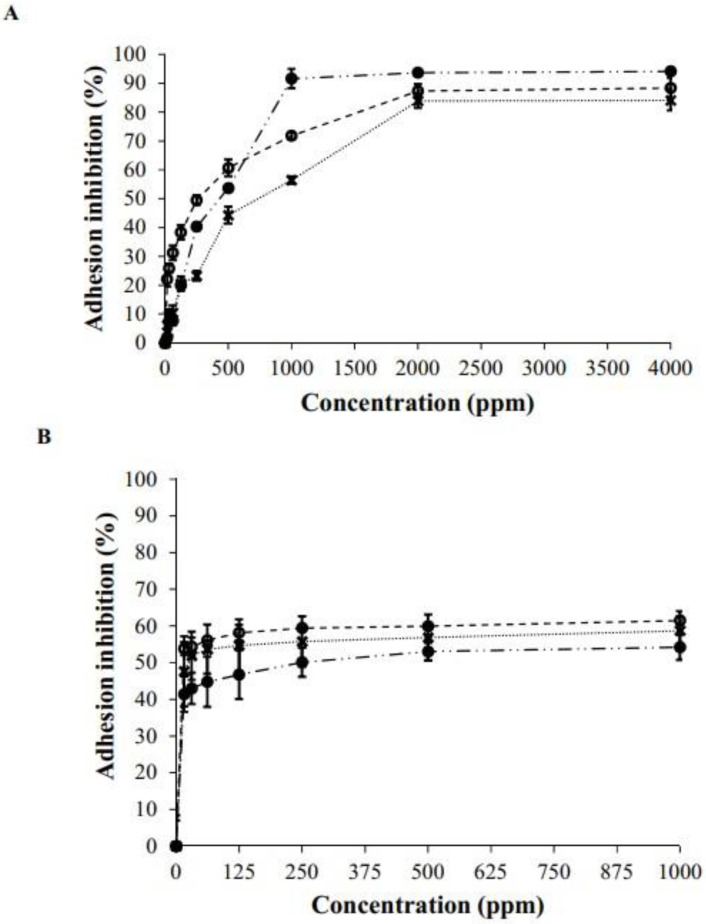
Adhesion inhibition response using BCE in pre-treatment (**A**) and post-treatment (**B**) against *Escherichia coli* (●), *Staphylococcus aureus* (X), and *Listeria monocytogenes* (○).

**Figure 5 microorganisms-10-01518-f005:**
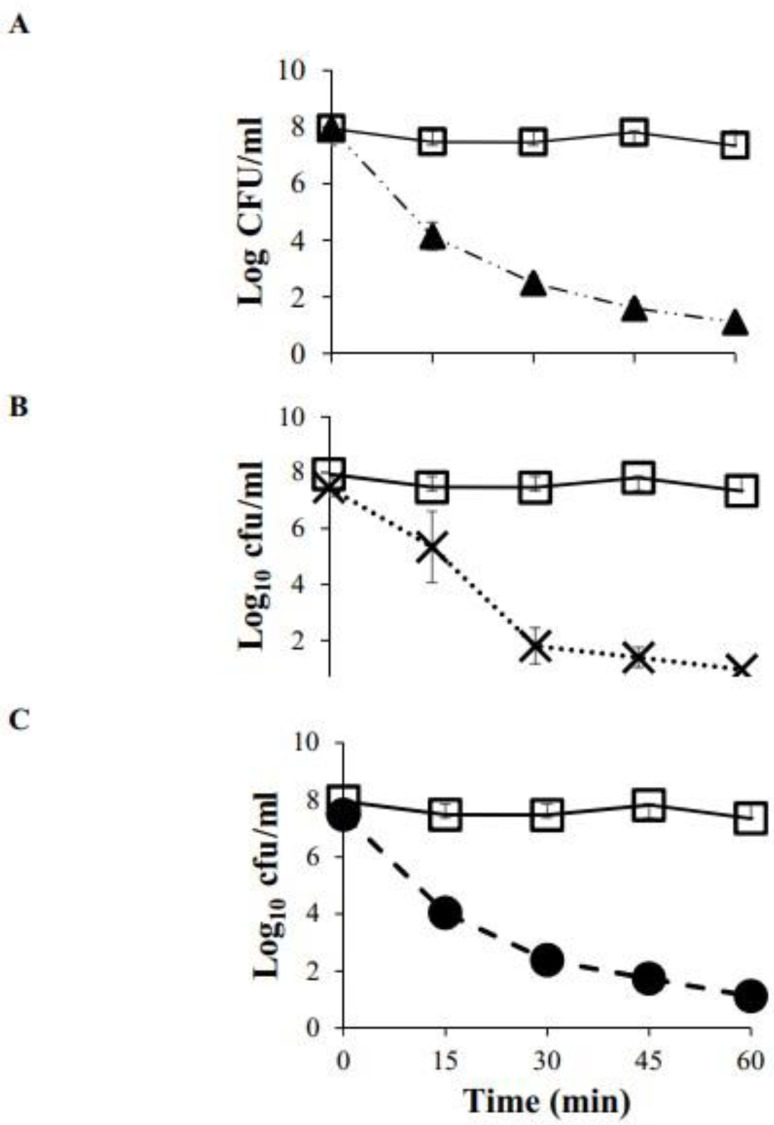
Reduction in cell viability (CFU/mL) versus time for (**A**) *Escherichia coli* (▲), (**B**) *Staphylococcus aureus* (X), (**C**) *Listeria monocytogenes* (●) and control (white square).

**Figure 6 microorganisms-10-01518-f006:**
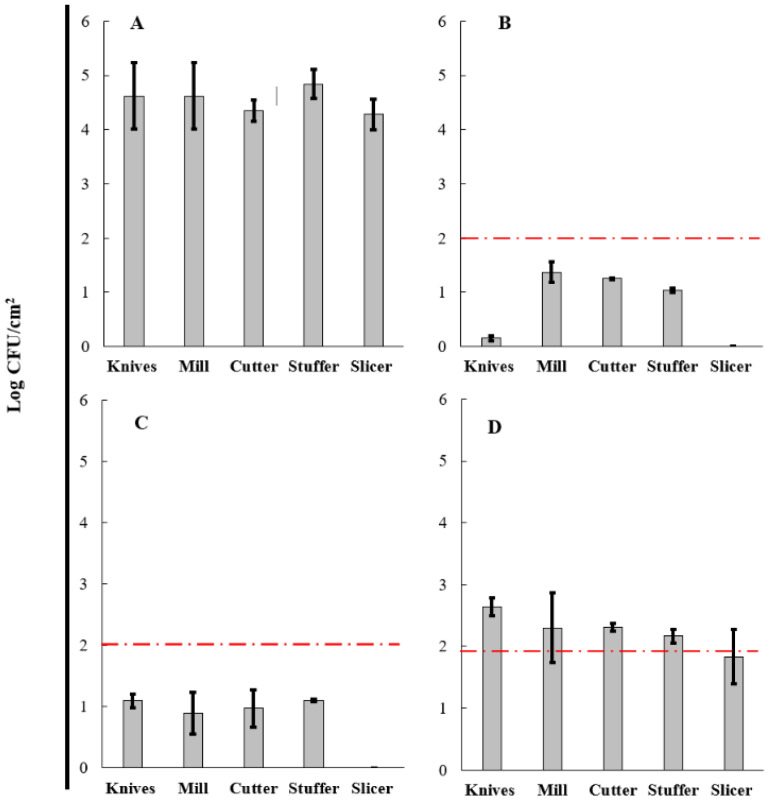
Aerobic mesophilic profile of knives, mill, cutter, stuffer, and slicer used during sausage production. Initial load before the cleaning process (**A**) and after the cleaning process using COP-A (**B**), COP-B (**C**), and COP-C (**D**). The limit established is shown in a red line.

**Figure 7 microorganisms-10-01518-f007:**
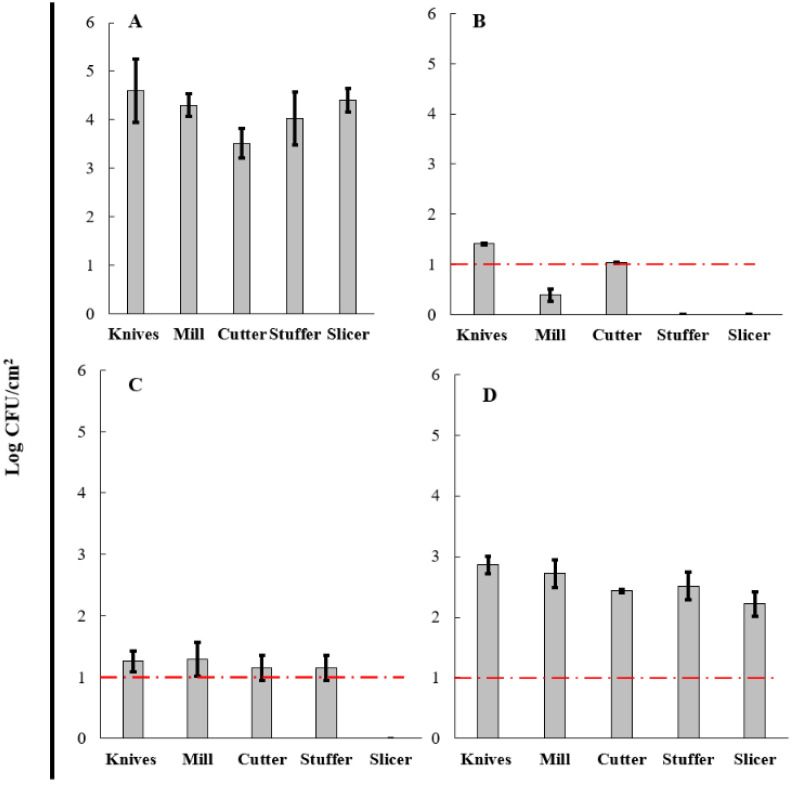
Total coliforms profile of knives, mill, cutter, stuffer, and slicer used during sausage production. Initial bacterial load before the cleaning process (**A**), and after the cleaning process using COP-A (**B**), COP-B (**C**), and COP-C (**D**). The limit established is shown in a red line.

**Table 1 microorganisms-10-01518-t001:** Sausage formulation produced in the meat processing laboratory.

Raw Material	% Composition
Lean pork	40
Lean beef	20
Fat	15
Pork leather	15
Isolated soy protein	5
Nitrites	1
Seasonings	4

**Table 2 microorganisms-10-01518-t002:** Percentage of microbial load reduction after the cleaning process of knives, mill, cutter, stuffer, and slicer used during sausage production.

Surfaces	(%) Reduction in Aerobic Mesophilic Microorganisms			(%) Reduction in Total Coliforms		
	COP-A	COP-B	COP-C	COP-A	COP-B	COP-C
Knives	96.75	76.13	42.94	66.88	70.45	33.02
Mill	69.13	79.95	48.08	90.95	69.91	36.81
Cutter	71.28	77.76	46.90	70.46	67.24	30.95
Stuffer	78.53	77.34	55.09	100.00	71.36	37.60
Slicer	100.00	100.00	57.35	100.00	100.00	49.55
Average reduction	83.14	82.24	50.07	85.66	75.79	35.58

## Data Availability

All data is reported in this article.

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
