# Peer review of "Biosurfactant from *Bacillus subtilis* DS03: Properties and Application in Cleaning Out Place System in a Pilot Sausages Processing"

_microorganisms, 2022, doi:10.3390/microorganisms10081518_

Round 1

Reviewer 1 Report

Dear Editor,

in this manuscript, Mendoza and co-authors described the sustainable production of biosurfactants from Bacillus subtilis DS03. In addition, the Authors characterized the biosurfactants by combining different assays including critical micelle concentration, emulsifying activity and antimicrobial activity.

Despite the large amount of experimental data, the manuscript presents the following criticisms:

The Authors did not perform high-resolution analysis (LC-MS, NMR) on BCE, therefore, its chemical nature is unknown. In my opinion, a high-resolution analysis is necessary to compare BCE with biosurfactants already known.

The BCE production should be performed by using sucrose as a carbon source. This control is fundamental to understand if the BCE production is related to the sucrose or to other elements present in molasses.

The assays on antimicrobial activity and cleaning-out-of-place (COP) process should be carried out by comparing BCE with a known biosurfactant with similar chemical features.

The comparison with SLES is not clear. Why did the Authors choose SLES?

Analytical points:

Line 98. What kind of molasses did the authors use? Beetroot or sugar cane?

Line 115. Why did the Authors treat cell supernatant with HCl up to pH 2?

Line 233. This paragraph should be better organized. At the present, the aims appear not clear and the discussion confused. Which is the major point to be addressed? I’ve noticed that the surface tension of BCE from whey (49 mN/m) is similar to that of Lactobacillus strains (41.90 mN/m). What is the reference value for classifying a substance as a biosurfactant?

Line 248. Sucrose and lactose are not polysaccharides!

Figure 2. Please add the growth medium in the figure caption.

Line 297. Please add more information about the extraction procedure. How did the Authors obtain three different fractions?

Line 303. Please add the results concerning the surface tension of BOE1 and BOE3.

Line 466. The composition of COP-A, B and C is described in the Material and Methods section; however here the differences between COP-A, B and C are not clear. Please add a brief description of these solutions. Moreover, this nomenclature creates confusion in the reader.

Figure 5. Standard deviations in full triangles and dots are not visible.

Author Response

Dear Reviewer,

We appreciate the information you provided, and we have considered each recommendation. Our answers to your concerns you can see in the document attached.

Regards,

Jonathan Colonel Leon

Reviewer 2 Report

Manuscript is well written. Some remarks:

Lines 224 – 230 – unnecessary repetition. The same text is above.

Lines 250 – 251 – please check this sentence “…and a slight increment in BCE production (0.85 ± 0.05 g/L) was observed compared to the control (0.618 ± 0.04).” – The values in brackets are for biomass – not for BCE

Figure 2 – BCE unit – here is mg/L. Is that correct for sure? In Figure 1 and in the text you have g/L

Author Response

(The authors gave the same response as above.)

Reviewer 3 Report

The manuscript written by Mendoza et al. is an ineresting study of Bacillus Biourfactant properties. The article is well written, and I have only few suggestion which can help to improve the text:

1) Fig. 3 could be bigger, especially 3A. 

2) Fig. 3C and 3D - I think that the horizontal axis should be logarythmic. Then the first part of the curve will be linear and Rwill be closer to 1. 

3) Fig. 4B - it is hard to distinguish the ponts at low concentrations. Maybe the chart should be divided into two - the first one general and the second one with zoom on the smaller concetrations?

4) Lines 428-452: I have met the CFU values presenthed like 10CFU/mL then 7 log CFU/mL. Please consider changing it.

5) Table 2: It is hard to judge about importance of results because there is no indication of statistical importance of differences between measurements. The standard deviation should be included at least. Then, the results description and conlusions (reffering to Table 2) should be reconsidered.

Author Response

(The authors gave the same response as above.)

Reviewer 4 Report

Dear Authors,

Your manuscript shows a very detailed and well designed set of experiments conducted not only in your labs but also at a pilot plant for meat production, which adds a great value to your report.

There are some minor grammar issues that I would suggest looking into. 

It would be very interesting to see a possibility to purify the crude BS extract and try to determine what kind of biosurfactant it might be (most probably it will be surfactin, but it should be confirmed by the chemical analyses). 

Since Bacillus spp. are known for producing some heat stable toxins, I would like to know if you had a chance to estimate their levels in your BCE. These are usually lipopeptides that have good emulsifying ability and therefore can affect your measurements of EI24. I would expand a discussion on it in point 3.2.2.

It would be also very interesting to see some form of discussion on using biosurfactants from sporing/pathogenic organisms for food industry. Maybe just a few sentences what the current guidance is and how to effectively ensure the safety around this matter. This however, is up to the Authors to consider and I am not forcing this point to be included. 

All in all, I think it is a very good manuscript and I hope to see more following. 

Author Response

(The authors gave the same response as above.)

Round 2

Reviewer 1 Report

Dear Editor,

the authors have partially responded to my requests. In my opinion, the manuscript is not yet suitable for publication.

-   - Although the BCE characterized in this work has application in the food industry, its chemical composition is essential. Now it is not known whether the effects described in this work are due to a glycolipid, a peptide, or a fatty acid or to a mixture of different compounds. In my opinion this work needs at least rough chemical characterization (chemical classes) of the BCE.

-         - I reiterate that the growth of Bacillus spp. on sucrose is an essential control for this work. The Authors should reproduce the experiments reported in Figure 1 using sucrose as a carbon source.

-  - The comparison between BCE and know biosurfactants used in this application is mandatory to understand its efficacy. In addition, the BCE comparison should be carried out in the same experimental conditions.

-         - The use of SLES should be explain in the text.

-       -   The COP nomenclature still unclear. Again, the COP nomenclature should be explained in the results section.